# Examining the Performance of Two Extraction Solvent Systems on Phenolic Constituents from U.S. Southeastern Blackberries

**DOI:** 10.3390/molecules26134001

**Published:** 2021-06-30

**Authors:** Xiaoxi Liao, Phillip Greenspan, Ronald B. Pegg

**Affiliations:** 1Department of Food Science & Technology, College of Agricultural and Environmental Sciences, The University of Georgia, 100 Cedar Street, Athens, GA 30602, USA; xiao.liao25@uga.edu; 2Department of Pharmaceutical & Biomedical Sciences, College of Pharmacy, The University of Georgia, 250 W. Green Street, Athens, GA 30602, USA; greenspn@uga.edu

**Keywords:** blackberries, phenolics, anthocyanins, ellagitannins, antioxidant activity, Caco-2 cells

## Abstract

Two common extraction solvent systems, namely acidified aqueous methanol and acidified aqueous acetone, were used to extract blackberry phenolics, and the antioxidant properties of the recovered extracts were compared. The crude extracts were fractionated into low- and high-molecular-weight phenolics by Sephadex LH-20 column chromatography. The hydrophilic-oxygen radical absorbance capacity (H-ORAC_FL_), ferric reducing antioxidant power (FRAP), and the cellular antioxidant activity (CAA) assays were employed as indices to assess antioxidant capacity of the extracts and their respective fractions. The methanolic solvent system displayed a greater efficiency at extracting anthocyanin and flavonol constituents from the blackberries, while the acetonic solvent system was better at extracting flavan-3-ols and tannins. Anthocyanins were the dominant phenolic class found in the blackberries with 138.7 ± 9.8 mg C3G eq./100 g f.w. when using methanol as the extractant and 114.6 ± 3.4 mg C3G eq./100 g f.w. when using acetone. In terms of overall antioxidant capacity of blackberry phenolics, the acetonic solvent system was superior. Though present only as a small percentage of the total phenolics in each crude extract, the flavan-3-ols (42.37 ± 2.44 and 51.44 ± 3.15 mg/100 g f.w. in MLF and ALF, respectively) and ellagitannins (5.15 ± 0.78 and 9.31 ± 0.63 mg/100 g f.w. in MHF and AHF, respectively) appear to account for the differences in the observed antioxidant activity between the two solvent systems.

## 1. Introduction

Worldwide, blackberries (*Rubus* spp.) are cultivated commercially and are a prominent source of polyphenols with purported antioxidant benefits. Two dominant classes of phenolics in blackberries are anthocyanins and hydrolyzable tannins [1,2]. Other phenolics are present and include phenolic acids, flavan-3-ols, and flavonols such as quercetin, kaempferol, isorhamnetin and their glycosylated derivatives [3,4]. Anthocyanins are colorful *O*-glycosidic pigments with a flavylium cation that have been long recognized for their health-promoting potential. Hydrolyzable tannins, notably ellagitannins, can be depolymerized to yield smaller phenolic compounds such as ellagic acid or hexahydroxydiphenoyl (HHDP) moieties, which were originally connected to a central glucose molecule via ester linkages, thereby contributing to the diversity in the extent of polymerization [5,6]. The dominant unit in blackberries is bis-HHDP-glucopyranose and its galloylated form, namely galloyl-bis-HHDP glucopyranose; the latter can be considered as the basic polymeric unit in many high-molecular-weight ellagitannins like lambertianin C [5].

To study these phytochemicals both qualitatively and quantitatively, they need to be extracted/isolated from the berry matrix using an appropriate solvent system. Subbiah et al. [1] stated that a multitude of organic solvents could be employed to extract phenolic compounds from berries, which could afterwards be identified and quantitated using different analytical methodologies like LC-ESI-QTOF-MS/MS. Many years earlier, Kähkönen et al. [7] reported that the most common solvent systems to extract phenolics from berries were aqueous mixtures of either ethanol, methanol or acetone. In their study, these researchers examined the efficacy of 70% aqueous acetone, 60% aqueous methanol, and pure water at extracting free phenolics from berries and apples. Based on preliminary data, aqueous acetonic extraction was chosen in preparing phenolic extracts from the set of twenty-eight berry and apple species because of its superior efficacy at extracting ellagitannins. The authors further noted that the addition of a small quantity (0.1–1%) of organic acid, for example, trifluoroacetic or formic acid, could further increase the yield of anthocyanins without causing any changes in acylated anthocyanin forms. Many extraction descriptions exist in the literature, but because of the diversity of antioxidant phytochemicals in botanicals like berries, there is no single extraction method that is considered optimal for all [8].

A number of factors are involved for ‘sufficient extraction’ of free phenolics from a food matrix and include, but are not limited to, the following: extent of maceration and particle size of the material; solvent combination employed and its polarity; solvent volume to sample mass ratio; temperature of extraction; duration of extraction; possible use of enzyme-, ultrasound-, or microwave-assistance to improve extractability; and the number of extractions carried out on the same material as a means to achieve an exhaustive extraction [9]. In reality, no one can validate whether all free phenolics have been completely recovered from a plant-based food. Moreover, there are also non-extractable or ‘bound’ phenolics endogenous to foods. Horticultural studies [10,11] have reported that phenolics in botanicals exist in free and covalently linked forms, such as esters and amides in cell wall material. To liberate these ‘bound’ phenolics prior to extraction, alkaline, acid, enzymatic hydrolysis, or combinations thereof are required [12]. Dr. Frank Sosulski’s lab characterized free, soluble-esterified, and insoluble-bound phenolic acids in oilseeds, cereal grains and potato flours [13,14,15]. The methodologies his team developed are still used today in phenolic analysis of foodstuffs, when bound phenolics need to be scrutinized [16]. An important question to consider is if the content of bound constituents should be taken into consideration when assessing the ‘phenolic profile’ of a food extract.

Limiting our discussion to hydrophilic phenolic constituents in a food matrix (as opposed to hydrophobic ones), the extractant chosen needs to be able to solubilize the phenolic classes present in the foodstuff. Early research tried to perfect a solvent system as the extractant. Water, methanol, ethanol, acetone, ethyl acetate and various combinations have been the usual solvents employed [17,18]. With regard to the alcohols, methanol is more frequently used than ethanol, because of its higher extraction efficiency [19]. It is important to note that we are discussing extractions from the point of view of chemical characterization of the polyphenolics present in foodstuffs, rather than preparing a food-grade decoction or dried extract for industrial purposes. Tsao and Deng [19] stated in their review that aqueous methanolic systems between 50 and 80% have been employed for extracting hydroxycinnamic acids, and many subgroups of flavonoids; a greater water composition in the solvent could aid in the extraction of glycosides of these compounds. In their review of phenolics in cereals, fruits and vegetables, Naczk and Shahidi [20] also commented on the extensively investigated use of methanol as an extractant with varying percentages of water added to it to increase the polarity. Recovered crude extracts were lyophilized, yields calculated, mass balances performed and the content of phenolics in the preparation then determined by quantitative HPLC analyses. Often the reason for choosing a particular solvent is not justified, and the completeness of extraction is never verified [21]. Hence, the publication of data from such studies can create issues when researchers try to compare/contrast their data against such published reports.

Bosso et al. [22] investigated the effectiveness of selected solvents, including water, ethanol, acetone and ethyl acetate, either neat or in combination, at extracting polyphenols from grape seed pomace. Aqueous acetone was deemed as the most effective extractant for procyanidin recovery, and the degree of polymerization was correlated with increased yield. Zhou and Yu [23] also reported a better recovery of polyphenolic compounds in wheat bran when using an acetonic solvent system. Contrarily, Naima et al. [24] compared the impact of different solvents on extraction yields from *Acacia mollissima* bark and reported that aqueous methanol was the best solvent to extract tannins and polyphenols. Acetone:water (70:30, *v*/*v*) has been reported to function as an excellent solvent system for the extraction of conjugated forms of ellagic acid in strawberries [25]. Another investigation conducted on strawberry phenolics showed that acidified methanol extracted a similar quantity of phenolics to that when acidified acetone was employed [26]. Using methanol with some added HCl for the recovery of anthocyanins has also been widely reported [27]. Yet, Garcia-Viguera et al. [28] found that acetonic extraction of strawberry anthocyanins yielded efficient, reproducible results, and could eliminate matrix issues due to the presence of pectin.

Even though green technologies for the extraction of bioactives from foodstuffs are being explored these days, the purpose of this study was to compare the extractability of the main phenolic classes found in a prominent U.S. Southeastern blackberry cultivar using the two most common solvent systems. As blackberries are rich in anthocyanins as well as ellagitannins, acidic aqueous methanolic and acetonic solvent systems were chosen as the extractants to be tested.

Many studies have reported the antioxidant capacities of plant polyphenols based on in vitro chemical assays. In this study, the crude phenolic extracts prepared from the methanolic and acetonic extraction systems were fractionated on a Sephadex LH-20 column into low- and high-molecular-weight constituents and then subjected to in vitro antioxidant assays; these included the hydrophilic-oxygen radical absorbance capacity (H-ORAC_FL_) and ferric reducing antioxidant power (FRAP) assays. To investigate the in vivo bioactive properties of dietary antioxidants, animal studies and human clinical trials are required; these are costly and time-consuming [29]. An alternative approach is to use a cell culture model as a means to analyze biological antioxidant properties. Such models factor in issues of antioxidant uptake, metabolism, and cell membrane permeability [30,31,32]. In this work, a cellular antioxidant activity (CAA) assay using the human colorectal adenocarcinoma (Caco-2) cell line was used to evaluate antioxidant activity in blackberry crude extracts from the methanolic and acetonic extraction systems. Finally, the phenolic constituents in the low- and high-molecular-weight fractions isolated by the Sephadex LH-20 column were identified and quantitated by HPLC–ESI–MS analyses.

## 2. Results and Discussion

### 2.1. Total Phenolics Content (TPC) and Antioxidant Capacities of the Phenolic Extracts Prepared from the Two Different Solvent Systems

The TPC values for the methanolic crude extract (MCE) and acetonic crude extract (ACE) were 371.1 ± 19.0 and 433.8 ± 15.5 mg gallic acid equivalents (GAE)/100 g fresh weight (f.w.) of blackberries, respectively (Table 1). These values are in the range of 292.2 to 446.4 mg/100 g f.w., as reported for U.S. Southeastern blackberry cultivars [33]. However, Toshima et al. [34] reported MCE contents of only ca. 75 and 110 mg GAE/100 g f.w. for ‘Kiowa’ and ‘Merton Thornless’ blackberries, respectively; ‘Kiowa’ blackberries are grown in the U.S. Southeast. The ACE exhibited a significantly (*p* < 0.05) greater TPC value than that of the methanolic counterpart (MCE).

Boeing et al. [35] also reported that blackberry phenolic extracts had greater TPC values when using aqueous acetone compared to an aqueous methanolic solvent: TPC values of 42.81 ± 0.28 and 27.74 ± 0.20 g GAE/kg dry weight were determined when a (70/30, *v*/*v*) solvent system of either aqueous acetone or aqueous methanol, respectively, was used. This finding for other foodstuffs or their byproducts is not uncommon. Recently, Kumar et al. [36] showed that aqueous acetonic extraction of peanut skins yielded the greatest amount of phenolics when compared to aqueous methanolic extracts. Mokrani and Madani [37] investigated the effects of solvent type (i.e., ethanol, methanol, acetone, and water) at various concentrations to extract the phenolic constituents of peach fruit. These researchers reported that amongst all the tested solvents, 60% acetone was the most efficient, according to TPC data, at extracting phenolics from peaches (363 mg GAE/100 g f.w.).

Significant differences in TPC values were observed also for the LMW and HMW fractions extracted using the two solvent systems: the methanolic solvent system extracted more LMW phenolics (MLF = 239.9 ± 4.8 mg GAE/100 g f.w.) compared to the acetonic system (ALF = 171.6 ± 4.5 mg GAE/100 g f.w.). On the other hand, the acetonic solvent system was more efficient at extracting HMW phenolics (121.0 ± 1.5 for the AHF vs. 60.9 ± 0.9 mg GAE/100 g f.w. for the MHF). The H-ORAC_FL_ value for the crude phenolic extract from ‘Ouachita’ blackberries was 4458 ± 508 μmol Trolox eq./100 g f.w. This value was similar to the 4100 μmol Trolox eq./100 g f.w. for the ‘Merton Thornless’ blackberry cultivar reported by Toshima et al. [34], but much greater than 2200 μmol Trolox eq./100 g f.w. stated in the same report for the ‘Kiowa’ cultivar. The H-ORAC_FL_ and FRAP values for the ACE and AHF were significantly (*p* < 0.05) greater than those of the MCE and MHF, respectively. To illustrate, the H-ORAC_FL_ values for the AHF and MHF were 1450 ± 70 vs. 477 ± 47 μmol Trolox eq./100 g f.w., while the respective FRAP values were 1113 ± 110 vs. 562 ± 36 μmol Fe^2+^ eq./100 g f.w. Differences in antioxidant activities between the two solvent extraction methods were not found to be significant (*p* > 0.05) for the LMW fractions. Even though the MLF possessed ca. a 40% greater phenolics content than its acetonic counterpart (i.e., ALF), as determined by the TPC assay, their H-ORAC_FL_ as well as FRAP values were not significantly (*p* > 0.05) different at 3498 ± 415 vs. 3645 ± 299 μmol Trolox eq./100 g f.w. and 1875 ± 101 vs. 1886 ± 17 μmol Fe^2+^ eq./100 g f.w., respectively.

The TMAC assay revealed a significant difference in terms of anthocyanin concentrations, by which the methanolic extraction system (145 ± 4.7 mg C3G eq./100 g f.w.) exhibited a better recovery than acetonic extraction (134 ± 3.1 mg C3G eq./100 g f.w.). Boeing et al. [35] reported a similar observation; however, their results were expressed on a dry weight basis: blackberry TMAC values of 7.50 ± 0.07 and 6.92 ± 0.01 g C3G eq./kg dry weight were determined when a (70/30, *v*/*v*) solvent system of either aqueous methanol or aqueous acetone, respectively, was used. On the other hand, Toshima et al. [34] reported markedly lower TMAC levels of ca. 40 and 50 mg C3G eq./100 g f.w. in the methanolic crude extract of ‘Kiowa’ and ‘Merton Thornless’ blackberry cultivars, respectively, from those of the present study.

### 2.2. Content of Major Phenolic Compounds from the Two Different Solvent Systems

HPLC–ESI–MS analysis revealed the presence of a number of phenolic compounds in the fractionated MLF, ALF, MHF, and AHF blackberry samples. Their concentrations were determined based on calibration curves. Table 2 lists four main phenolic classes, and the compounds identified in each for the LMW fractions from both the methanolic and acetonic extraction solvent systems employed. Ellagitannins were determined to be exclusively in the HMW fractions; so, their contents are also reported as a fifth class of phenolics in Table 2.

In the LMW fractions, anthocyanins were the dominant phenolic in blackberries, as determined by HPLC quantitative analysis. The contents (138.7 ± 9.8 mg C3G eq./100 g f.w. for MLF; 114.6 ± 3.4 mg C3G eq./100 g f.w. for ALF) are comparable to the values reported by Fan-Chiang and Wrolstad [38], where the total anthocyanin contents ranged from 75 to 201 mg C3G eq./100 g f.w. based on the analysis of thirty-nine Oregon blackberry varieties. Similar to the TMAC results, aqueous methanol afforded superior extraction of anthocyanins when compared to those in the acetonic extract. In agreement with our study, Chirinos et al. [27] revealed a higher level of anthocyanins from mashua tubers using 90% methanolic extraction rather than 90% acetonic extraction. In this work, the methanolic solvent system also demonstrated a greater recovery of flavonol glycosides (35.83 ± 2.93 mg/100 g f.w.) compared to 29.10 ± 1.83 mg/100 g f.w. in the acetonic extract. Galankis et al. [39] noted that more polar solvents, like hydro-alcoholic mixtures, were efficient in extracting flavonoid glycosides and higher molecular-weight phenols, which might explain the efficacy for the acidified aqueous methanolic extractant. Rupasinghe et al. [40] also reported that methanolic extraction yielded greater quantities of quercetin and its glycosides from apple peels than did an analogous acetonic system. The contents of phenolic acids in the MLF (2.97 ± 0.54 mg/100 g f.w.) and ALF (2.71 ± 0.32 mg/100 g f.w.) were similar, but overall they contributed an insignificant amount to the total phenolics in the blackberry LMW fractions. Flavan-3-ols including (epi)catechin-4,8′-(epi)catechin hexoside and propelargonidin B-type dimer were quantitated in the LMW fractions, with the latter being the dominant compound (from 60 to 64%) of the total flavan-3-ols. Noteworthy is that the total content of flavan-3-ols was greater in the ALF (51.44 ± 3.15 mg/100 g f.w.) than in the MLF (42.37 ± 2.44 mg/100 g f.w.), indicating that aqueous acetone was a more effective solvent at extracting flavan-3-ols. This observation has been documented in other agricultural products [41,42]. For instance, Chavan et al. [41] reported that the extraction efficiency of condensed tannins, based on (+)-catechin as the flavan-3-ol monomer, from different pea samples using acidified aqueous acetone (i.e., 70%, 80%, 90%, 100% solutions) vs. that of acidified aqueous methanol at the same concentrations was statistically greater (*p* < 0.05) at every concentration tested. For beach pea, the best extraction efficiencies of 11.6 ± 0.19 vs. 4.54 ± 0.67 g of condensed tannin/100 g meal were determined for 70% acidified acetonic and methanolic extractants, respectively. Moreover, in all experiments, the addition of 1% HCl to the acetonic and methanolic extractants afforded superior extraction efficacies than counterpart solvent systems devoid of the acid. As Galanakis et al. [39] pointed out; the tendency of each phenol class to be solubilized, transferred, or diffused into a given solvent system is governed by thermodynamics, in particular, the activity coefficient of the phenolics in question. The stereochemistry of the phenolics (i.e., the polar and non-polar moieties within a molecule) and the intermolecular forces, like hydrogen bonding, between them and the solvent system will dictate the extent of solubility.

Table 2 also reports on the major ellagitannins identified and quantitated in the HMW blackberry fractions. The total ellagitannins content in AHF (9.31 ± 0.63 mg/100 g f.w.) was nearly twice that of the MHF (5.15 ± 0.78 mg/100 g f.w). Additionally, the H-ORAC_FL_ and FRAP values given in Table 1 for the AHF were two to three times of those for the MHF, indicating that phenolic contents were positively associated with measured in vitro antioxidant capacities. Noteworthy is that the yield of LMW phenolics accounted for the lion’s share of the total phenolics extracted (i.e., ~95%) with the methanolic solvent system being superior (Table 2). The extracted ellagitannins in the MHF and AHF accounted for only a small percentage of the total phenolics; yet, they were largely responsible for the superior antioxidant capacity of the acetonic extract.

### 2.3. Cellular Antioxidant Activity (CAA) Assay

In this study, CAA measurements were assessed against a quercetin standard, because it afforded a better response than other antioxidant standards tested by Wolfe and Liu [31]. In the assay, the fluorescence signal increases as the fluorescent probe, DCFH, is oxidized by peroxyl radicals arising from the added free-radical generator, ABAP. The presence of antioxidants in the system can quench free radicals, thereby preventing them from attacking the DCFH probe. Thus, Caco-2 cells not treated with a phenolic extract/fraction will exhibit greater fluorescence. Cells treated with quercetin followed a dose-dependent inhibition of fluorescence (14.3 to 30.1%) with increasing quercetin concentrations ranging from 25 to 200 μM (Figure 1). The inhibition plateaued at the highest concentration of 200 μM; that is, no further reduction in fluorescence was observed at higher concentrations (data not shown). To demonstrate differences between the two extraction methods, phenolics were added to cells on the basis of f.w. of blackberries. Cells treated with 25 mg of blackberry f.w. eq./mL showed greater inhibition of fluorescence compared to the cells treated with only 10 mg of blackberry f.w. eq./mL (Figure 2).

Significantly (*p* < 0.05) greater inhibition of cellular oxidation was observed for the ACE treatment than that of MCE at equi-concentrations on a f.w. basis (Figure 2), which further validates the postulate that greater inhibition is due to the elevated levels of extracted ellagitannins and procyanidins in the acetonic preparation. The superior antioxidant activity of these phenolics might be attributed to a greater number of hydroxy groups in the polymeric chemical structures of the tannin constituents [43]. Moreover, the structures of phenolic compounds are closely related to antioxidant activity: the 3-hydroxy group between the 2,3-double bond in the A-ring and the 3′,4′-*O*-dihydroxy moiety in the B-ring, such as that found in quercetin and flavan-3-ols, exhibited the greatest antioxidant activity in the CAA assay [44]. McDougall et al. [45] studied the polyphenol compositions of berry extracts and their antiproliferative effectiveness on human cervical and Caco-2 cells. These researchers demonstrated that the observed antiproliferative activity was attributed chiefly to the ellagitannins and procyanidins, rather than the anthocyanins. Furthermore, Reddy et al. [46] investigated the antioxidant activity of isolated phenolic compounds from pomegranate. They found that ellagitannins were superior in their antioxidant activity compared to anthocyanins. The greater antioxidant activity observed in this work for the acetonic extract is believed to arise from the slightly higher quantities of the flavan-3-ols and ellagitannins recovered from blackberries using this solvent system. Our CAA findings correspond to the data from the in vitro antioxidant capacity assays, where the acetonic extract displayed greater antioxidant activity (Table 1). In contrast, Wolfe and Liu [44], examining a variety of flavonoids using HepG2 cells, reported that the data from H-ORAC_FL_ were not associated with CAA data. Yet, Wan et al. [47] performed the experiment using Caco-2 cells and did find a correlation between CAA and H-ORAC_FL_ values of rat plasma after the rodents had consumed antioxidants. The apparent difference may possibly be due to active membrane transport systems of the two different cell lines, which could be composed of different uptake or efflux transporters [48,49,50]. The possible uptake and efflux mechanisms for phenolics endogenous to blackberries have not been extensively studied.

## 3. Materials and Methods

Figure 3 summarizes the sample preparation and assays performed on the blackberry crude phenolic extracts and their fractions.

### 3.1. Chemicals

ACS-grade solvents (acetone, methanol, and 95% ethanol), HPLC-grade solvents (water, methanol, and acetonitrile), glass wool, Gibco fetal bovine serum, glacial acetic acid and hydrochloric acid were acquired from the Fisher Scientific Co., LLC (Suwanee, GA, USA). Sephadex LH-20, Amberlite XAD-16, Folin & Ciocalteu’s phenol reagent, gallic acid (>99%), (+)-catechin hydrate (>99%), chlorogenic acid (≥98%), protocatechuic acid (≥97%), vanillic acid (≥97%), fluorescein 3′,6′-dihydroxyspiro[isobenzofuran-1[3H]9′[9[H]xanthen]-3-one), Trolox [(±)-6-hydroxy-2,5,7,8-tetramethylchroman-2-carboxylic acid, 97%], TPTZ (2,4,6-tripyridyl-*S*-triazine, ≥99.0%), quercetin-3-*O*-rutinoside hydrate (≥94%), iron(II) sulfate heptahydrate (≥99.0%), iron(III) chloride hexahydrate (97%), 2′,7′-dichlorofluorescin diacetate (DCFH-DA, ≥97%), and 2,2′-azobis [2-amidinopropane] dihydrochloride (ABAP) were purchased from the Sigma-Aldrich Chemical Company (St. Louis, MO, USA). Cyanidin-3-*O*-glucoside chloride was obtained from the Indofine Chemical Company, Inc. (Hillsborough, NJ, USA). Advanced DMEM, phosphate-buffered saline (PBS), Hanks’ balanced salt solution (HBSS), penicillin-streptomycin, and trypsin-EDTA were purchased from Life Technologies (Grand Island, NY, USA). Human colorectal adenocarcinoma (Caco-2) cells were procured from the American Type Culture Collection (ATCC, Manassas, VA, USA).

### 3.2. Blackberry Sample Preparation

Thornless blackberries (*Rubus fruticosus* ‘Ouachita’ PP17162) were handpicked at Jacob W. Paulk Farms, Inc. (Wray, GA, USA) over two summers. This cultivar features an erect-type morphology and was released by the University of Arkansas breeding program. Six lots of blackberries (~1200 g per lot) were collected each summer, vacuum packaged (Henkelman 600, Henkelman BV, The Netherlands) and stored at −40 °C. Representative samples (~150 g × 3) were randomly taken from all lots, the berries lyophilized using a FreeZone 2.5 L freeze dryer (Labconco Corporation, Kansas City, MO, USA), and then stored in a −80 °C freezer until analysis.

### 3.3. Extraction of Blackberry Phenolics

Lyophilized blackberries were ground to a fine powder using a Ninja NJ100 Express Chop (SharkNinja, Newton, MA, USA). Phenolics were extracted from the sample using two common solvent systems: (CH_3_OH/H_2_O/HCl, 70.0/29.0/1.0, *v*/*v*/*v*) or ((CH_3_)_2_CO/H_2_O/CH_3_COOH, 70.0/29.5/0.5, *v*/*v*/*v*). Briefly, blackberry powder (15 g) was mixed with one of the solvent systems at a material-to-solvent ratio of 1:10 (*w*/*v*). The mixture was blended with a PT-3100 Polytron™ homogenizer (Brinkmann Instruments, Westbury, NY, USA) at 13,000 rpm for 10 min, and centrifuged at 4000 rpm for 20 min. The extract was then filtered through Whatman No.1 filter paper (Whatman International Ltd., Maidstone, England) using a Büchner funnel. The extraction was repeated twice as described above, and the filtrates were pooled. A Rotavapor R-210 connected to a V-700 vacuum pump with a V-850 vacuum controller (Büchi Corporation, New Castle, DE, USA) was used to evaporate the organic solvent under reduced pressure at 45 °C. The remaining aqueous fraction was subjected to simple sugar and organic acid removal following a method reported by Srivastava et al. [51]. The Amberlite XAD16 column was washed with deionized water until the eluent achieved a 0.0% Brix reading when measured with a digital handheld refractometer (PAL-1; Atago USA, Inc., Bellevue, WA, USA). Methanol was then applied to the column to recover the phenolic extract. The methanol was removed using the Rotavor. The aqueous residue in the recovered methanolic and acetonic crude phenolic extracts (abbreviated as MCE and ACE, respectively) was then removed by freeze-drying, and the lyophilized MCE or ACE stored in amber-glass vials at 4 °C. The above extraction was performed in triplicate for each solvent system.

### 3.4. Sephadex LH-20 Column Chromatography

Low- and a high-molecular-weight (LMW and HMW) fractions from the MCE and ACE were collected using a Sephadex LH-20 chromatographic column [52]. A crude extract (200 mg) was dissolved in 95% (*v*/*v*) ethanol and loaded onto the column. Flow through the column was initiated by gravity using 95% (*v*/*v*) ethanol to elute the LMW fraction. The mobile phase was then changed over to 50% (*v*/*v*) aqueous acetone to recover the HMW fraction. Solvents were evaporated from collected fractions using the Rotavapor at 45 °C. The remaining aqueous residue was removed by lyophilization, and the product then stored in amber-glass vials at 4 °C. The prepared methanolic LMW and HMW fractions as well as acetonic LMW and HMW fractions are abbreviated as MLF, MHF, ALF, and AHF, respectively.

### 3.5. Total Phenolics Content (TPC) and Antioxidant Assays

The phenolic concentrations and antioxidant capacities of the blackberry extracts and fractions were assessed via the TPC, H-ORAC_FL_, and FRAP assays described by Robbins et al. [52]. The TPC, H-ORAC_FL_, and FRAP values were expressed as mg GAE/100 g f.w., μmol Trolox eq./100 g f.w., and μmol Fe^2+^ eq./100 g f.w, respectively. Triplicate measurements were carried out for each assay.

### 3.6. Total Monomeric Anthocyanin Content (TMAC) Assay

The anthocyanins concentrations of the crude phenolic extracts prepared using the two different solvent extraction systems (i.e., MCE and ACE) were determined using the pH differential method described by Giusti and Wrolstad [53]. The assay takes advantage of the natural characteristic of anthocyanins in acidic condition: they exist in colored oxonium-ion and colorless hemiketal forms at pHs of 1.0 and 4.5, respectively. The extracts were dissolved in two buffer systems: 0.025 M potassium chloride buffer at pH 1.0, and 0.4 M sodium acetate buffer at pH 4.5, at concentrations ranging from 25 μg/mL to 100 μg/mL. The absorbance was read at both wavelengths of 510 nm and 700 nm, the latter correcting for any light scattering arising from turbidity in the solutions. The assays were performed in triplicate and the TMAC values were calculated using the formulas below:*A* = [(*A*_510_ − *A*_700_)_pH 1.0_ − (*A*510 − *A*_700_)_pH 4.5_](1)
TMAC (mg C3G eq./L) = (*A* × MW × DF × 1000)/(ε *×* ℓ)(2)
where *A* = absorbance; MW = molecular weight (449.2 g/mol); DF = dilution factor; and ε = molar extinction coefficient of cyanidin-3-*O*-glucoside (26,900 L cm^−1^ mol^−1^). TMAC data for the crude phenolic extracts were then expressed as mg C3G eq./100 g f.w.

### 3.7. HPLC–Electrospray Ionization–Mass Spectrometry (HPLC–ESI–MS) Characterization

Following the method reported by Gong and Pegg [54], characterization and quantitation of the phenolics in the MLF, ALF, MHF, and AHF samples were conducted using an Agilent 1200 series HPLC coupled with a reversed-phase Kinetex^®^ XB-C18 column (150 × 4.6 mm *i.d.*, 2.6-μm particle size, 100 Å; Phenomenex, Torrance, CA, USA). A linear gradient elution of mobile phase A (H_2_O/CH_3_CN/CH_3_COOH, 93:5:2, *v*/*v*/*v*) and B (H_2_O/CH_3_CN/CH_3_COOH, 58:40:2, *v*/*v*/*v*) from 0% to 80% B over a 35 min period at a flow rate of 0.6 mL/min was used. The MLF & ALF as well as MHF & AHF samples were dissolved in methanol at 1 and 1.5 mg/mL, respectively, and passed through a 0.2-μm regenerated cellulose syringe filter. The injection volume was 20 μL. The wavelengths for detection were set at λ = 255 nm (ellagic acid and its derivatives), 280 nm (benzoic acid family, flavan-3-ols), 330 nm (hydroxycinnamic acids), 360 nm (flavonols), and 520 nm (anthocyanins). Tentative identification was made by matching UV–Vis spectra and retention time (*t*_R_) mapping with authentic standards.

Confirmation of separated phenolics was conducted on an Agilent 1100 HPLC system coupled to a Waters^®^ QToF micro™ Mass Spectrometer in the negative-ion mode with an ESI interface (Waters Corporation, Milford, MA, USA). The capillary voltage was −2.5 kV, the nitrogen flow rate was 450 L/h, and the ion-transfer capillary temperature was at 300 °C. Argon was used as the collision gas. The collision voltages were set at 5 V for MS and 20 V for MS/MS. The detection was set in a scanning mass range of 80 to 3000 Da. Identifications were made by comparing parent molecular ions [M-H]^−^ and fragmentation patterns against those of known standards and reported literature values. The calculated phenolic contents in the MLF, ALF, MHF, and AHF samples were expressed as mg/100 g f.w. of blackberries.

### 3.8. Cellular Antioxidant Activity (CAA) Assay

Cellular antioxidant measurements were carried out following the procedure employed by Kellett et al. [50]. Advanced DMEM medium supplemented with 10% endotoxin-free, heat-inactivated fetal bovine serum (FBS), 1% L-glutamine and 1% penicillin-streptomycin were used as the culture media. The quercetin standard was prepared in the concentration range from 25 to 200 μM. Both MCE and ACE were dissolved in 95% (*v*/*v*) ethanol and tested at concentrations of 10 and 25 mg of blackberry f.w. eq./mL. The cells were cultured in a Corning Costar^®^ 96-well plate and incubated at 37 °C with 5% (*v*/*v*) CO_2_ until confluence. On the day of analysis, the media was removed and the cells were washed three times with PBS. The working DCFH-DA solution (50 μL) was added, followed by 50 μL of the culture media containing the quercetin standard, or a blackberry phenolic extract, or blanks (i.e., culture media containing the equivalent amount of ethanol). The cells were incubated at 37 °C for 60 min. After incubation, the media was removed, and cells were washed again with PBS (3×). One hundred microliters of 600 μM ABAP prepared in HBSS were added to each treatment well as a free-radical generator, and the plate was immediately transferred to the BMG FLUOstar Omega microplate reader (BMG LABTECH Inc., Cary, NC, USA) for measurement. The excitation/emission wavelength was set at 485/538 nm for fluorescence signal detection. A total of 13 cycles of fluorescent response was collected and the area under the curve (AUC) was calculated using MARS Data Analysis Software (BMG LABTECH). A reduction in fluorescence was calculated using the equation:(3)$$\%\;\mathrm{reduction}=(1-\frac{A_{s}}{A_{b}})\;*\;100$$
where, A_s_ and A_b_ referred to AUC of samples and blank, respectively.

### 3.9. Statistical Analysis

For HPLC analysis, each phenolic extract and fraction was analyzed on two different days over a period of a week. For each experiment, two injections were performed; hence, quadruplicate measurements were obtained. Interday measurements were performed for the CAA assay on two different days. When each assay was carried out, data were collected in quadruplicate for each treatment giving *n* = 8. A comparison between the means for TPC, antioxidant assays, and CAA measurements were established by the least square means procedure and Tukey’s multiple range test using the JMP Pro software, Version 13 (Cary, NC, USA). Statistical significance was considered at *p* < 0.05.

## 4. Conclusions

In the scientific literature, one can find various solvent systems and sample preparation approaches employed to extract phenolic compounds from foodstuffs. It is obvious that solubilization of phenolics is a complicated process, and its efficiency is governed by a complex interplay of parameters. Assuming that the approaches described are sufficiently detailed and performed to the best ability of the researchers, the results reported should be considered acceptable. Clearly, however, there are limitations in what the data signify, based on the methodology employed. With so many variations in the methods/approaches used to extract phenolics, it is difficult to make true and appropriate comparisons of reported data. The present investigation highlights this very fact. We employed the two most common extractants in phenolic research for the recovery of phenolics from U.S. Southeastern ‘Ouachita’ blackberries. The results indicate that when choosing a solvent system, one must be selected that will extract/isolate the main phenolics of interest for further examination. In this study, though the methanolic system was superior in recovering anthocyanins (138.7 ± 9.8 vs. 114.6 ± 3.4 mg C3G eq./100 g f.w. in the MLF and ALF, respectively) and flavonols, the acetonic extract was found to possess more flavan-3-ols and ellagitannins. More ellagitannins present in the AHF (i.e., 5.15 ± 0.78 vs. 9.31 ± 0.63 mg/100 g f.w. in the MHF and AHF, respectively) were found to have a greater impact on the measured antioxidant indices, according to both in vitro chemical assays as well as the CAA. When tannin constituents are major phenolics of concern in a foodstuff, such as blackberries, this study highlights that aqueous acetone as the extractant likely will afford better recovery of free phenolics contributing to the overall antioxidant capacity found in blackberries.

## Figures and Tables

**Figure 1 molecules-26-04001-f001:**
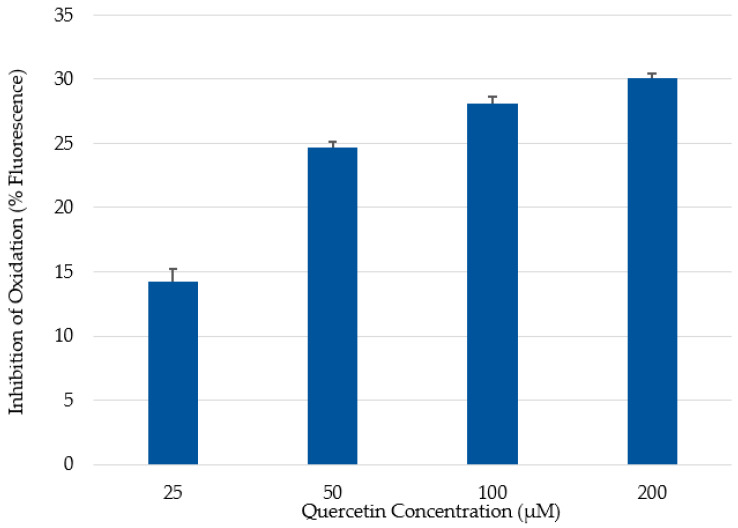
Cellular antioxidant activity (CAA) of quercetin at concentrations ranging from 25 to 200 μM.

**Figure 2 molecules-26-04001-f002:**
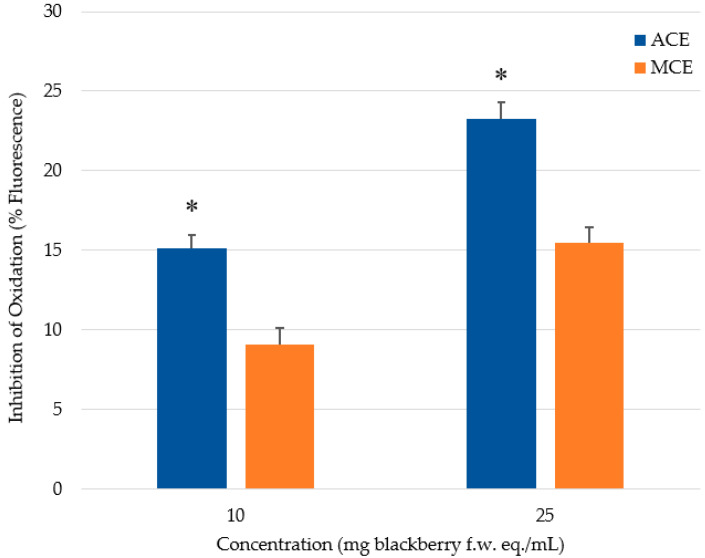
Antioxidant activity of phenolic crude extracts in the CAA assay. Extracts were added to cells on the basis of mg of blackberry fresh weight (f.w.) equivalent (eq.) per mL. * denotes significant (*p* < 0.0001) difference when compared to methanol extracts at the same concentration.

**Figure 3 molecules-26-04001-f003:**
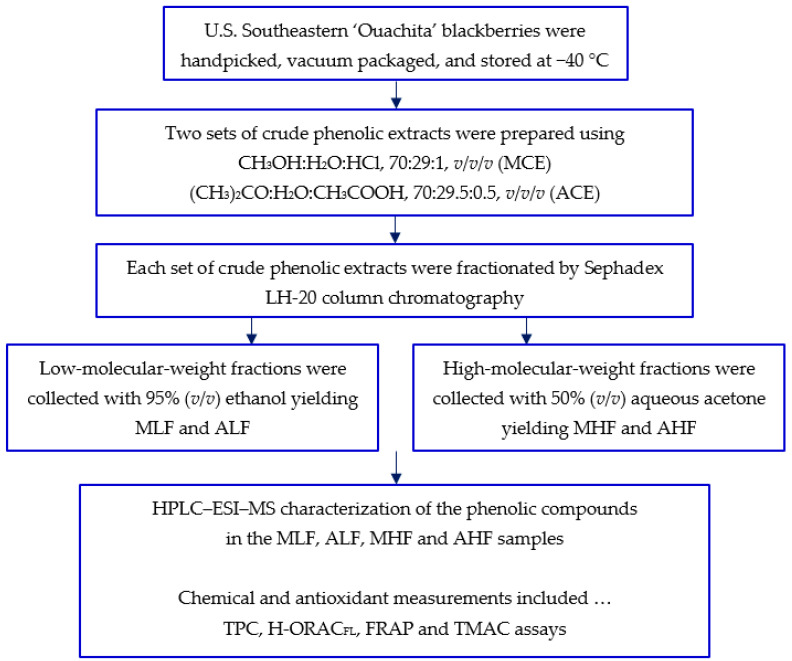
Flow diagram outlining the sample preparation and assays performed on the blackberry crude phenolic extracts and their fractions.

**Table 1 molecules-26-04001-t001:** Total phenolics content (TPC), antioxidant capacity determinations, and total monomeric anthocyanin content (TMAC) of blackberry samples extracted using two different solvent systems ^1^.

Samples ^2^	TPC (mg GAE/100 g f.w.) ^3^	H-ORAC_FL_ (μmol Trolox eq./100 g f.w.) ^4^	FRAP (μmol Fe^2+^ eq./100 g f.w.) ^5^	TMAC (mg C3G eq./100 g f.w.) ^6^
methanol:water:hydrochloric acid (70.0/29.0/1.0, *v*/*v*/*v*) extraction				
MCE	371.1 ± 19.0 b	4458 ± 508 b	2538 ± 150 b	145 ± 4.7 b
MLF	239.9 ± 4.8 b	3498 ± 415 a	1875 ± 101 a	–
MHF	60.9 ± 0.9 b	477 ± 47 b	562 ± 36 b	–
acetone:water:acetic acid (70.0/29.5/0.5, *v*/*v*/*v*) extraction				
ACE	433.8 ± 15.5 a	6529 ± 560 a	3403 ± 372 a	134 ± 3.1 a
ALF	171.6 ± 4.5 a	3645 ± 299 a	1886 ± 17 a	–
AHF	121.0 ± 1.5 a	1450 ± 70 a	1113 ± 110 a	–

^1^ Values for the crude methanolic extract and its fractions in each column with the same letter to the corresponding acetonic extract are not significantly (*p* > 0.05) different, as determined by Tukey’s multiple range test. All data are reported as means ± standard deviations (*n* = 9). ^2^ Abbreviations are as follows: LMW, low-molecular-weight; HMW, high-molecular-weight; MCE, methanolic crude extract; MLF, methanolic LMW fraction; MHF, methanolic HMW fraction; ACE, acetonic crude extract; ALF, acetonic LMW fraction; and AHF, acetonic HMW fraction. ^3^ TPC, total phenolics content; GAE, gallic acid equivalents; and f.w., fresh weight blackberries. ^4^ H-ORAC_FL_, hydrophilic-oxygen radical absorbance capacity; eq., equivalents; and f.w., fresh weight blackberries. ^5^ FRAP, ferric reducing antioxidant power; eq., equivalents; and f.w., fresh weight blackberries. ^6^ TMAC, total monomeric anthocyanins content; C3G eq., cyanidin-3-*O*-glucoside equivalents; and f.w., fresh weight blackberries.

**Table 2 molecules-26-04001-t002:** Quantitative determinations of dominant phenolic classes (mg/100 g f.w.) found in Georgia-grown ‘Ouachita’ blackberries based on the extraction solvent system employed ^1^.

Phenolic Compounds Identified by HPLC–ESI–MS ^2^	Extraction Solvent System	
	Methanolic	Acetonic
	Phenolic acids in MLF/ALF	
protocatechuic acid hexoside	0.57 ± 0.06	0.55 ± 0.03
*p*-coumaric acid derivative	0.60 ± 0.06	0.55 ± 0.10
hydroxybenzoic acid hexoside	0.39 ± 0.02	0.35 ± 0.03
ellagic acid derivative	1.41 ± 0.40	1.25 ± 0.16
Total	2.97 ± 0.54	2.71 ± 0.32
	Flavan-3-ols in MLF/ALF	
(epi)catechin-4,8′-(epi)catechin hexoside	15.26 ± 0.88	19.64 ± 0.82
propelargonidin B-type dimer	27.11 ± 1.56	31.80 ± 2.33
Total	42.37 ± 2.44	51.44 ± 3.15
	Anthocyanins in MLF/ALF	
cyanidin-3-*O*-glucoside	122.1 ± 7.4	99.88 ± 2.24
cyanidin-3-*O*-arabinoside	7.31 ± 1.20	6.83 ± 0.41
cyanidin derivative	5.69 ± 0.70	4.43 ± 0.45
cyanidin-3-*O-*(6″-dioxalylglucoside)	3.57 ± 0.49	3.22 ± 0.31
Total	138.7 ± 9.8	114.4 ± 3.4
	Flavonols in MLF/ALF	
isorhamnetin derivative	5.58 ± 0.32	4.62 ± 0.37
quercetin-3-*O*-rutinoside	4.62 ± 0.65	4.39 ± 0.52
quercetin-3-*O*-galactoside	7.68 ± 0.73	7.06 ± 0.14
quercetin-3-*O*-glucoside	4.25 ± 0.20	4.02 ± 0.02
quercetin derivative	5.10 ± 0.48	4.98 ± 0.39
quercetin derivative	5.16 ± 0.35	2.39 ± 0.22
quercetin-3-*O*-acetylglucoside	3.44 ± 0.20	1.64 ± 0.17
Total	35.83 ± 2.93	29.10 ± 1.83
	Ellagitannins in MHF/AHF	
castalagin	0.60 ± 0.10	1.47 ± 0.09
lambertianin C isomer	2.90 ± 0.50	5.47 ± 0.41
sanguiin H-6	1.65 ± 0.18	2.37 ± 0.13
Total	5.15 ± 0.78	9.31 ± 0.63

^1^ Quantitation was performed using a Kinetex^®^ XB-C18 column (4.6 × 150 mm *i.d.*, 2.6-μm particle size, 100 Å; Phenomenex), and the data are reported as mg equivalents (eq.) of the available standard/100 g f.w. Commercial standards included protocatechuic acid, vanillic acid, (+)-catechin hydrate, cyanidin-3-*O*-glucoside chloride, and quercetin-3-*O*-rutinoside hydrate. The data are reported as means ± standard deviations (*n* = 3). ^2^ HPLC–ESI–MS, high-performance liquid chromatography–electrospray ionization–mass spectrometry.

## Data Availability

Not applicable.

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
