# Peer review of "Examining the Performance of Two Extraction Solvent Systems on Phenolic Constituents from U.S. Southeastern Blackberries"

_molecules, 2021, doi:10.3390/molecules26134001_

Round 1
Reviewer 1 Report
Dear Author(s)
After an exhaustive revision, the manuscript is Reconsider after major revision (control missing in some experiments). In general, the study is closely connected to the journal's objectives. The study is very interesting. However, the authors need to make some changes and modify parts in the manuscript, mainly in the section “Results and Discussion”.
In the following pages, I give a detailed revision of the manuscript.
Best regards
General comments
** The authors must add the initials of the authors **
ABSTRACT
The abstract is good. Only, the authors need to add the main results in terms of numbers, %, among others.
- INTRODUCTION
General comments
The introduction is good, since it starts from general to particular. The English is good.
The introduction needs to clarify some details and update the references. Specifically, the authors should search references from this year, since the authors not added studies from 2021. Update the references will allow the authors to enhance the introduction with more current data, as well as other sections of the study.
I recommend these papers:
- https://www.mdpi.com/1420-3049/26/4/1187/htm
- https://www.mdpi.com/1420-3049/26/2/327/htm
- https://www.mdpi.com/1420-3049/26/8/2146/htm
My observations:
"… using an appropriate solvent system"
The authors should add lines with reason(s) that indicate the importance of a solvent extraction system.
“Many extraction descriptions exist in the literature, but there is no single extraction method that is considered standard.”
The authors need to add the types of extractions.
“An important question to consider is if the content of bound constituents should be taken into consideration when assessing the ‘phenolic profile’ of a food extract.”
These lines are confusing. What is the meaning of these lines?. What is(are) the reference(s) for these lines?
“Limiting our discussion to hydrophilic phenolic constituents in a food matrix (as op-posed to hydrophobic ones), the extractant chosen needs to be able to solubilize the phe-nolic classes present in the foodstuff. Early research tried to perfect a solvent system as the extractant. Water, methanol, ethanol, acetone, ethyl acetate and various combinations have been the usual solvents employed. For instance, methanol has been extensively investigated as an extractant with varying percentages of water added to it to increase the polarity. Recovered crude extracts were lyophilized, yields calculated, mass balances performed and the content of phenolics in the preparation then determined by quantitative HPLC analyses. Often the reason for choosing a particular solvent is not justified, and the completeness of extraction is never verified. Hence, the publication of data from such studies can create issues when researchers try to compare/contrast their data against such published reports.”
The authors indicate many results. What are the references for these lines?
“For instance, methanol has been extensively investigated as an extractant with varying percentages of water added to it to increase the polarity. Recovered crude extracts were lyophilized, yields calculated, mass balances performed and the content of phenolics in the preparation then determined by quantitative HPLC analyses.”
The authors need to add the results from these lines, since the lines are very generic, it is necessary to add some numerical data, with the reference.
“Because blackberries are rich in anthocyanins as well as ellagitannins, both acidic methanolic and aqueous acetonic solvent systems were employed in the present work to compare the extractability of phenolic classes from the berries.”
The reviewer recommends to the authors that the objective of the study be written at the end of the introduction (with the other part of the objective). Thus, the objective is clearer to understand.
- RESULTS AND DISCUSSION
General comments
"Results and Discussion" is characterized by a description of the results, the explication of the results, comparison with other studies and explication (discussion) of the results obtained with respect to other studies.
My observations:
2.1. Total phenolics content (TPC) and antioxidant capacities of the phenolic extracts
The description of the results is very complete, very detailed, which is good. However, the authors not present any explication of the results. The comparison is very poor “Boeing et al. [20] reported a similar finding to the one reported here.”, it is important to add more details and explication (discussion) of the results obtained with respect to other studies.
2.2. Content of Major Phenolic Compounds from the Two Different Solvent Systems
The authors need to add the standard deviation in each result.
Why in Table from 2.1 appears MCE, MLF, MHF, ACE, ALF, and AHF and in Table from 2.2 only methanolic and acetonic? The authors need to add all the information in each point, since the information is easier to understand.
The description of results is very short, but very concise and precise. However, the authors not present any explication of the results. The comparison with other studies is good. However, the authors not present any explication (discussion) of the results obtained with respect to other studies.
2.3. Cellular Antioxidant Activity (CAA) Assay}
This subsection is very complete. The description of results is very short, but very concise and precise, with explication of results. The comparison with other studies is good and discussion is complete. However, why appears “methanolic and acetonic” in Figure 2? The Figure should show MCE, MLF, MHF, ACE, ALF, and AHF, similar to 2.1 and 2.2.
- MATERIALS AND METHODS
General comments
This section is clear. The English is good. The authors must add a Figure that represents all the methodology in the section Materials and Methods. This Figure will help to understand the methodology.
My observations:
3.5. Total Phenolics Content (TPC) and Antioxidant Assays
What are the units of measurement? What is the standard?
3.6. Total Monomeric Anthocyanins Content (TMAC) Assay
What are the units of measurement? What is the standard?
3.7. HPLC–Electrospray Ionization–Mass Spectrometry (HPLC–ESI–MS) Characterization
What are the units of measurement?
- CONCLUSIONS
The conclusions should be improved from the changes made in the manuscript.
REFERENCES
The authors follow the author's guide for references.
Reviewer 2 Report
Present research by Liao et al. is focused on evaluation of polyphenols content and bioactivity of blackberries extracts. Aqueous methanol and acetone were evaluated as extraction solvents while extracts were characterized in terms of spectrophotometric assays (total phenols and in vitro antioxidant activity), quantitative HPLC–ESI–MS for polyphenols determination and cellular antioxidant activity. Text is well written and experiments seem to be very carefully performed. The topic should be of interest for the Molecules readers. However, I have some doubts about the novelty of the paper. There have been numerous studies evaluating blackberries and blackberry pomace chemical profile and bioactivity. Even, novel extraction techniques were applied for this purpose.
- Therefore, I suggest that authors should highlight the novelty of this work and step forward comparing to previously published research.
- Highlight the application of these results in the Conclusions section.
- Did you consider application of green solvents for this issue and how do you compare that with acetone and methanol?
Round 2
Reviewer 1 Report
Dear Author(s)
After an exhaustive revision, the manuscript is Accept in present form. The resubmitted manuscript has been completely improved compared to its previous version. Therefore, the manuscript can be published in “Molecules”.
Best regards.
Reviewer 2 Report
Paper is improved but lack of novelty in approach is my only concern.